

# Diverse methanogens, bacteria and tannase genes in the feces of the endangered volcano rabbit (*Romerolagus diazi*)

Leslie M. Montes-Carreto[1], José Luis Aguirre-Noyola[2], Itzel A. Solís-García[3], Jorge Ortega[4], Esperanza Martinez-Romero[2] and José Antonio Guerrero[1]

[1] Facultad de Ciencias Biológicas, Universidad Autónoma del Estado de Morelos, Cuernavaca, Morelos, Mexico
[2] Centro de Ciencias Genómicas, Universidad Nacional Autónoma de Mexico, Cuernavaca, Morelos, Mexico
[3] Red de Estudios Moleculares Avanzados, Instituto de Ecología, A.C., Xalapa, Veracruz, Mexico
[4] Escuela Nacional de Ciencias Biológicas, Instituto Politécnico Nacional, Ciudad de Mexico, Mexico

Corresponding authors
Esperanza Martinez-Romero,
emartine@ccg.unam.mx
José Antonio Guerrero,
aguerrero@uaem.mx

## ABSTRACT

**Background**. The volcano rabbit is the smallest lagomorph in Mexico, it is monotypic and endemic to the Trans-Mexican Volcanic Belt. It is classified as endangered by Mexican legislation and as critically endangered by the IUCN, in the Red List. *Romerolagus diazi* consumes large amounts of grasses, seedlings, shrubs, and trees. Pines and oaks contain tannins that can be toxic to the organisms which consume them. The volcano rabbit microbiota may be rich in bacteria capable of degrading fiber and phenolic compounds.

**Methods**. We obtained the fecal microbiome of three adults and one young rabbit collected in Coajomulco, Morelos, Mexico. Taxonomic assignments and gene annotation revealed the possible roles of different bacteria in the rabbit gut. We searched for sequences encoding tannase enzymes and enzymes associated with digestion of plant fibers such as cellulose and hemicellulose.

**Results**. The most representative phyla within the Bacteria domain were: Proteobacteria, Firmicutes and Actinobacteria for the young rabbit sample (S1) and adult rabbit sample (S2), which was the only sample not confirmed by sequencing to correspond to the volcano rabbit. Firmicutes, Actinobacteria and Cyanobacteria were found in adult rabbit samples S3 and S4. The most abundant phylum within the Archaea domain was Euryarchaeota. The most abundant genera of the Bacteria domain were *Lachnoclostridium* (Firmicutes) and *Acinetobacter* (Proteobacteria), while *Methanosarcina* predominated from the Archaea. In addition, the potential functions of metagenomic sequences were identified, which include carbohydrate and amino acid metabolism. We obtained genes encoding enzymes for plant fiber degradation such as endo 1,4 $\beta$-xylanases, arabinofuranosidases, endoglucanases and $\beta$-glucosidases. We also found 18 bacterial tannase sequences.

## INTRODUCTION

Mammals establish symbiotic relationships with microorganisms that colonize different regions of the body, such as skin, mucous membranes, gastrointestinal tract, among others (*Muegge et al., 2011*). Microbiota is defined as a microbial community living in an anatomical site, self-regulating their abundance and metabolic dynamics, which can influence the health status of the host. Microbiome is the term used to designate the genome of the microbiota (*Redinbo, 2014*; *Ventura et al., 2009*). Herbivores have been characterized as having a larger microbial diversity in their guts than omnivores or carnivores (*Ley et al., 2008*). Bacterial symbionts expand the host digestion spectrum by fermenting cellulose and hemicellulose (*Ley et al., 2008*; *Tilg, 2010*). Individual/intrinsic factors that includes age, sex, genetics, host phylogeny and environmental/extrinsic factors like diet and habitat conditions could change the microbial community of the rabbit gut (*Funosas et al., 2021*; *North, Zotte & Hoffman, 2019*).

The rabbit gastrointestinal tract presents a 6−6.5 pH, associated with high humidity (75–95%) and stable temperature of 35−40 °C. The transit speed of particles and food fluids is slower (27 and 39 h respectively) compared to other species as the guinea pig (13 h), rat (22 and 20 h) and horse (25 and 20 h) (*Velasco-Galilea et al., 2018*). The rabbit gut microbial community is composed of 100–1,000 billion microorganisms per gram of cecal, belonging to 1,000 different species of microorganisms (*Combes et al., 2011*). Bacteria belonging to Lachnospiraceae, Clostridiaceae, and Ruminococcaceae families play an important role in the digestion of cellulose and hemicellulose as they may produce short-chain fatty acids (*Biddle et al., 2013*). In addition, other families that include Desulfovibrionaceae, Eubacteriaceae, Bacteroidaceae, Christensenellaceae, Erysipelotrichaceaea, Rikenellaceae and Spirochaetaceae have been reported in lagomorphs (*North, Zotte & Hoffman, 2019*; *Shanmuganandam et al., 2020*; *Stalder et al., 2019*).

Diet is considered one of the factors modeling the microbial community in all animals and this is also the case with hindgut-fermenter animals like rabbits (*Muegge et al., 2011*). Moreover, rabbits are caecotrophagic animals (they ingest their own soft fecal pellets produced by digestion in the cecum) profiting from the nutrients in feces derived from microbial fermentation (*Crowley et al., 2017*). Fecal ingestion would allow a second digestion of plant fibers, a process that could be considered analogous to ruminant digestion. Coprophagy is as well a very efficient way to recycle gut microbiota (*Martinez-Romero et al., 2021*). In most lagomorph studies, Firmicutes were found as the most abundant bacteria (*North, Zotte & Hoffman, 2019*; *Stalder et al., 2019*; *Velasco-Galilea et al., 2018*) followed by Bacteroidetes (*North, Zotte & Hoffman, 2019*; *Stalder et al., 2019*), although *Crowley et al. (2017)* found that both Bacteroides and Firmicutes were equally abundant. Proteobacteria were found to be more abundant in cecal samples (*North, Zotte & Hoffman, 2019*), which differs from the results with fecal samples investigated in other papers (*Velasco-Galilea et al., 2018*; *Shanmuganandam et al., 2020*). It should be noted that Illumina 16S rRNA v3-v4 sequencing led to an underrepresentation of Firmicutes (*Shanmuganandam et al., 2020*). Firmicutes and Bacteroidetes are the dominant phyla in human guts (*Human Microbiome Project Consortium, 2012*) and therein the

Firmicutes/Bacteroidetes ratio is not constant and changes with age (*Mariat et al., 2009*) and in some cases with obesity (*Magne et al., 2020*).

The volcano rabbit (*Romerolagus diazi*) is an endemic species classified as endangered by Mexican legislation (*SEMARNAT, 2010*) and as critically endangered in the IUCN Red List (*Velázquez & Guerrero, 2019*). It plays an important ecological role as part of the diet of carnivorous mammals, prey for birds and reptiles (*Cervantes & Martínez-Vázquez, 1996*; *Uriostegui Velarde & García, 2015*). Besides, the volcano rabbit consumes large amounts of vegetative parts of grasses, seedlings, shrubs, trees and can regulate plant composition and seed dispersal processes (*Granados-Sánchez, López-Ríos & Hernández-García, 2004*). The diet of the volcano rabbit is based on grasses but is complemented with other plants. Previously, 37 plant species were identified in its diet, 80% of which were grasses such as *Muhlenbergia macroura*, *Festuca amplissima* and *Jarava ichu*. In addition, the consumption of leaves of *Phoradendron velutinum*, *Pinus* sp, and *Quercus laurina* was recorded (*Mancinez Arellano, 2017*). Pines and oaks contain chemical compounds that can be toxic to the organisms that consume them. These toxins are considered to have a defensive role against animals (*Granados-Sánchez, Ruíz-Puga & Barrera-Escorcia, 2008*). The aim of this study was to describe the microbes found in the volcano rabbit feces to assess their role in plant fiber and tannin degradation, because tannins are important constituents of their diet. We surmised that the fecal microbiome of the volcano rabbit would reflect its herbivorous diet as observed in other animals (*Martinez-Romero et al., 2021*) with a plethora of genes encoding enzymes to degrade plant fibers and phenolics.

## MATERIALS & METHODS

### Sample collection

Four fecal samples were collected from Coajomulco, Morelos Mexico (14Q UTM 478960.07 m E, 2109691.32 m N) under a permit by Secretaría de Medio Ambiente y Recursos Naturales (SGPA/DGVS/006985/18). These pellets are ochre in color, with a smooth, shiny texture and their maximum diameter is one centimeter in adults. We kept more than 25 m between one latrine and another within a collection site, this distance is the estimated home range for these rabbits (*Cervantes & Martínez-Vázquez, 1996*), except for two samples (S1 and S2) that were collected in the same latrine with 0.1 m distance between them, but the pellets were of different sizes. Feces were stored individually in Eppendorf-type tubes in an icebox and later taken to the laboratory and stored in a refrigerator at 4 °C for 6 h.

### DNA extraction, sequencing, and Illumina library generation

Total DNA of the four samples was extracted using the commercial DNA Isolation Kit from Roche Life Science. A total of 230 mg of feces were processed for each sample. DNA concentration was measured by Nanodrop (Thermo Scientific) and Qubit™ dsDNA HS Assay Kits and the quality was determined by visualization in a 1% agarose gel. The sample requirements for the preparation of the libraries were: DNA concentration > 200 ng/μL, total amount of DNA > 1 μg and DNA bands visualized in agarose had to be clear and of good quality. The genomic DNA was randomly cut into small fragments and subcloned into a "universal" cloning vector. The sub fragment library was randomly sampled, and

several sequences reads (using a universal primer that directs sequencing from within the vector) were generated from both strands. Sequencing was obtained with Illumina HiSeq 4000 using paired-end 2× 151 bp, at the sequencing unit Functional Genomics Laboratory at UC Berkeley, California.

## Sequence filtering and metagenome assembly

Raw fastq reads were quality filtered with FASTP 0.20.0 (*Chen et al., 2018*). We considered good quality reads when the quality score was equal to or greater than 30. Clean reads were mapped against genomes such as human, rat, rabbit, yeast, mouse, pine, oak, fly, worm, adapters, and pika to know whether the reads had contamination using FastQ Screen 0.14.0 (*Wingett & Andrews, 2018*). Data is available in the SRA at NCBI, BioProject: PRJNA721235, accession number: SRR14209496, SRR14209495, SRR14209494, SRR14209493). Metagenome assembly was performed using metaSPADES 3.12.0 (*Nurk et al., 2017*) and MEGAHIT 1.1.3 (*Li et al., 2015*). Comparison of the assemblies was conducted to identify the best assembly with MetaQUAST 5.0.2 (*Mikheenko, Saveliev & Gurevich, 2016*). A good quality assembly considers the proportion of contigs <1000 nt, total length of the contigs, longest contig, N50 (length), L50 (position), fewer contigs and identifying single-copy genes (>90%) (*Nurk et al., 2013*). Clean reads (R1 and R2) were aligned to each corresponding assembly to obtain the percentage of reads that were assembled using Bowtie2 2.4.2 (*Langmead & Salzberg, 2012*). We confirmed that the samples were from volcano rabbits because we encountered DNA sequences in feces that matched those reported from the same rabbit species using the nucleotide sequences of the cytochrome *b* gene. We assessed taxonomic identity on fecal samples using Blast 2.10.0+ (*Camacho et al., 2009*) and a phylogenetic reconstruction using maximum likelihood (ML). Additional sequences of related organisms were obtained by Blast 2.10.0+ at NCBI. Sequences were aligned with Mafft 7.149b (*Katoh & Standley, 2013*). The best substitution model (TPM2+F+G4) and ML analysis were performed with IQ-TREE 1.6.12 (*Kalyaanamoorthy et al., 2017*; *Trifinopoulos et al., 2016*), using the UFBoot2 method (*Hoang et al., 2018*) with 10,000 replicates.

## Taxonomic Assignment

The microbial taxonomy assignation of each assembly per sample was estimated using Kraken 2 2.0.8 (*Wood, Lu & Langmead, 2019*). The abundance of the microbial community by phylum and genus was estimated using Bracken 2.5.0 (Bayesian Abundance Re-estimation with KrakEN). Bracken could reassign sequences that kraken 2 could not classify within the genus or species levels with a reliable percentage above 98% (*Lu et al., 2017*). Both programs were run against a Minikraken2_v1 database containing bacterial, archaeal genomes and viral libraries. In addition, we searched for the 16S rRNA gene sequences in the metagenomes by Blast 2.10.0+. Redundant sequences were removed with CD-HIT 4.8.1 (*Li & Godzik, 2006*). Phylogenetic reconstruction was performed using the maximum likelihood (ML) method. Additional sequences of related organisms were obtained by Blast at NCBI and with the Refseq-RDP database. Sequences were aligned with Mafft 7.149b. Spurious bases, editing and trimming were performed with trimAl

1.4 (*Capella-Gutierrez, Silla-Martinez & Gabaldon, 2009*). TThe best substitution model (GTR+F+R5) and ML analysis were performed with IQ-TREE 1.6.12 (*Kalyaanamoorthy et al., 2017*; *Trifinopoulos et al., 2016*), using the UFBoot2 method (*Hoang et al., 2018*) with 10,000 replicates.

### Alpha diversity and microbial composition

The microbial diversity of each sample at both phylum and genus level of Bacteria and Archaea was estimated with the Hill numbers in terms of effective numbers of elements under the same sample coverage (*Chao & Jost, 2012*). Hill numbers are a mathematically unified family of diversity indices differing only by an exponent $q$ ($^qD$) (*Chiu & Chao, 2016*), where $q = 0$ is equivalent to species richness, $q = 1$ corresponds to the exponential of Shannon entropy (effective number of common elements), and $q = 2$ is equivalent to the inverse of Simpson index, interpreted as the effective number of dominant elements (*Ma & Li, 2018*; *Ma, Li & Gotelli, 2019*). To compare the microbial diversity between the four samples, we used the 95% confidence intervals (CI), where no overlap between CI values indicates significant differences (*Cumming, Fidler & Vaux, 2007*). The qD diversity, sample coverage, and their respective confidence intervals were obtained with the iNEXT R package 3.5.3 (*Hsieh et al., 2016*), using as endpoint the maximum number of contigs in each sample and 1,000 bootstraps for the construction of the rarefaction curves and CI.

The difference in composition of bacterial and archaeal communities was assessed by one-way ANOSIM based on Bray-Curtis similarity metrics and with 10,000 permutations. When the ANOSIM was significant, a pairwise comparison was conducted between samples using the Bonferroni *p*-value correction as implemented in Past 4.2 (*Hammer, Harper & Ryan, 2001*). This analysis was conducted considering only those genera with relative abundance equal to or greater than 0.5%, meaning that rarely occurring microorganisms were not considered.

### Predicted functional annotation

Gene annotation and coding sequence identification in feces microbiomes were performed using Prokka 1.12 (*Seemann, 2014*). One step for cleaning the protein fasta file to eliminate redundant sequences was performed using CD-HIT. We used blastp 2.10.0+ on the sequences that Prokka did not score (hypothetical proteins) against the UniProtKB/Swiss-Prot database. Additionally, non-redundant protein fasta files of each sample were obtained. We performed a second annotation of protein sequences with the online program GhostKOALA 2.2 (*Kanehisa, Sato & Morishima, 2016*), using GHOSTX search (it uses suffix arrays to find matching sequences and runs 100 times faster than BLAST), against a non-redundant set of KEGG genes (Kyoto Encyclopedia of Genes and Genomes).

## RESULTS

### Sequence and metagenome assembly

We extracted DNA from four fecal samples. According to the size of the collected feces, S1 (0.5 cm) corresponded to a young individual, S2, S3 and S4 (1.0 cm) corresponded to adult rabbits (*Velázquez, Romero & López-Paniagua, 1996*). We confirmed that the

feces samples S1, S3, and S4 were from volcano rabbits because we found cytochrome *b* sequences in feces matched those reported previously from the same rabbit species (*Osuna et al., 2020*). Although cytochrome *b* sequence was not detected from sample S2, we are confident that it corresponds to volcano rabbit as feces of other rabbits inhabiting the zone are clearly distinctive (*Velázquez, Romero & López-Paniagua, 1996*). Maximum likelihood analysis indicated that the cytochrome *b* gene sequences found in the metagenomes were phylogenetically placed with the volcano rabbit (Fig. S1).

We obtained a total of 332.8 million raw reads from the four samples. Once the cleaning was completed, we rescued 316.6 million reads and the average number of reads per sample was 79.15 million (ranging from 69.6 to 96.6) (Table S1). Extra filtering of the clean sequences was conducted to obtain external contamination. The results demonstrated that between 98.2% and 99.6% of the clean reads were not assigned to adapters, human, rat, rabbit, yeast, mouse, pine, oak, fly, worm, and pika genomes. Individual assemblies of each sample S1, S2, S3 and S4 were performed. Fewer and larger contigs were obtained with metaSPADES than with MEGAHIT. Moreover, 83%–85% of the clean reads were mapped against each assembly.

## Taxonomic assignment

The percentage of contigs that could be classified ranged between 14% and 18% for the taxonomic levels of phylum and genus of the Bacteria and Archaea domains. This could be due to the limitations of the non-human database where rare species are present (*Tamames, Cobo-Simon & Puente-Sanchez, 2019*). Twenty-nine phyla were obtained from the Bacteria domain (Fig. 1A). The phylum Proteobacteria was the most abundant in sample two (S2) and sample from the young rabbit (S1) that has been found in other young animals and humans (*Moon et al., 2018*). The most abundant phylum in samples S3 and S4 was Firmicutes. Additionally, we found three phyla of the Archaea domain: Euryarchaeota, Crenarchaeota and Thaumarchaeota with the most abundant being Euryarchaeota (Fig. 1B). Nine families were the most abundant in all samples (Table 1). The most abundant genus of the domain Bacteria in the young rabbit feces was *Acinetobacter* (Proteobacteria) while in adult-rabbit feces was *Lachnoclostridium* (Firmicutes) (Fig. 2A). *Candidatus tachikawaea*, *Leclercia*, *Candidatus riesia*, and *Obesumbacterium* were only found in the gut metagenome from young rabbit feces (Fig. 2A). The number of archaeal genera we encountered here in the fecal microbiome of the volcano rabbit is remarkable. We identified 15 genera of the domain Archaea, *Methanosarcina*, *Methanoculleus* and *Methanococcus* were the most abundant in all samples. The genus *Halorabdus* was the most abundant only in sample two (Fig. 2B), which is the only sample for which there was no confirmation of host rabbit identity.

We recovered 169 16S rRNA sequences (1,273 ~1,500 bp) from the assembled metagenomes. Maximum likelihood analysis indicated that the 16S rRNA gene sequences found in the metagenomes were phylogenetically placed in the following phyla: Firmicutes, Bacteroidetes, Proteobacteria, Synergistetes, Tenericutes, Verrucomicrobia, Cyanobacteria, Lentisphaerae and Spirochaetes (Fig. S2). Besides, we found some 16S rRNA gene sequences that did not cluster with any reported phyla. Many 16S rRNA gene sequences were related

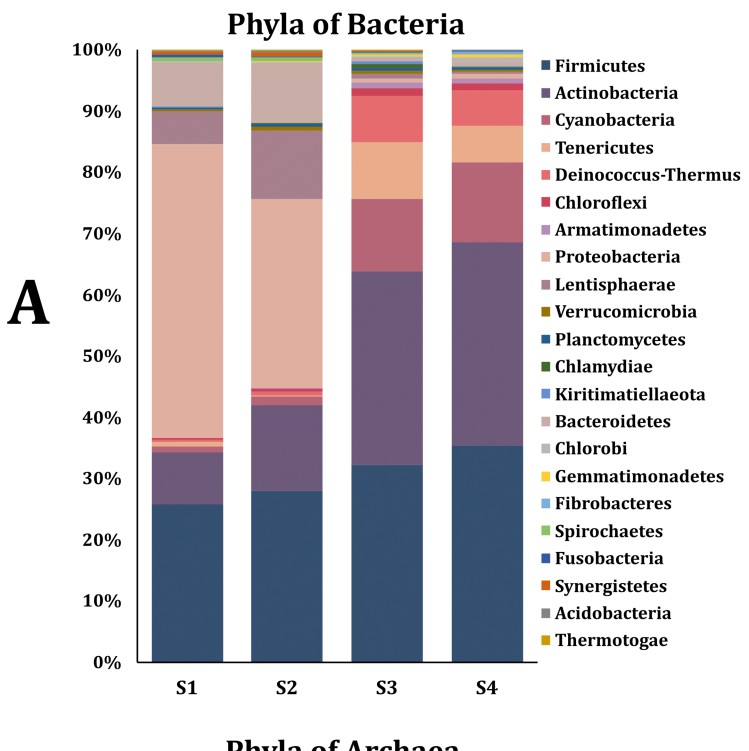

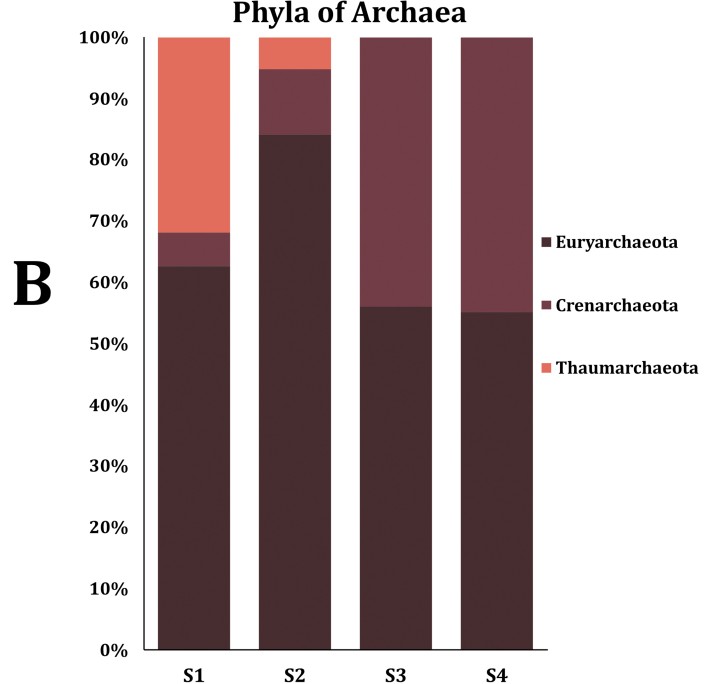

**Figure 1** **Phyla of the bacteria and archaea domain found in the fecal microbiome of the volcano rabbit.** Relative abundance was obtained by dividing the number of contigs assigned per phylum by the total number of contigs for the Bacteria and Archaea domains, respectively. The figures were constructed considering only phyla with a relative abundance equal to or greater than 0.5%, which means that rare bacteria and archaea were not considered. (A) Percentage of abundance of the main phyla from the Bacteria domain found in feces of the volcano rabbit. (B) Percentage of abundance of the main phyla from the Archaea domain found in feces of the volcano rabbit.

Table 1 Most abundant phyla, families, and genera in the fecal microbiomes of the volcano rabbit.

| Phylum | Family | Genus | Global relative abundance of the genus (%) |
|---|---|---|---|
| Proteobacteria | Moraxellaceae | *Acinetobacter* | 2.82 |
| | | *Moraxella* | 1.25 |
| | | *Psychrobacter* | 1.23 |
| | Pseudomonadaceae | *Azotobacter* | 0.87 |
| | | *Pseudomonas* | 1.09 |
| | Hafniaceae | *Hafnia* | 0.24 |
| | | *Obesumbacterium* | 0.20 |
| | | *Edwardsiella* | 0.26 |
| | Erwiniaceae | *Erwinia* | 0.23 |
| | | *Pantoea* | 0.24 |
| | Yersiniaceae | *Serratia* | 0.22 |
| | Enterobacteriaceae | *Enterobacter* | 0.70 |
| | | *Citrobacter* | 0.67 |
| | | *Klebsiella* | 0.65 |
| | | *Escherichia* | 0.51 |
| | | *Lelliottia* | 0.48 |
| | | *Salmonella* | 0.50 |
| | | *Cronobacter* | 0.46 |
| | | *Kosakonia* | 0.38 |
| | | *Raoultella* | 0.31 |
| | | *Pluralibacter* | 0.35 |
| | | *Cedecea* | 0.29 |
| | | *Leclercia* | 0.25 |
| | | *Candidatus Riesia* | 0.25 |
| Firmicutes | Lachnospiraceae | *Lachnoclostridium* | 5.59 |
| | | *Blautia* | 3.02 |
| | | *Roseburia* | 3.45 |
| | | *Butyrivibrio* | 2.61 |
| | | *Herbinix* | 1.60 |
| | | *Cellulosilyticum* | 1.39 |
| | | *Anaerostipes* | 2.04 |
| | | *Lachnoanaerobaculum* | 1.83 |
| | Clostridiaceae | *Clostridium* | 1.23 |
| | | *Mordavella* | 1.16 |
| | | *Geosporobacter* | 1.02 |
| | | *Alkaliphilus* | 1.12 |
| | | *C. Arthromitus* | 1.00 |

**Table 1** (*continued*)

| Phylum | Family | Genus | Global relative abundance of the genus (%) |
|---|---|---|---|
| | Oscillospiraceae | *Ruminococcus* | 1.00 |
| | | *Faecalibacterium* | 1.08 |
| | | *Oscillibacter* | 0.74 |
| | | *Flavonifractor* | 1.13 |
| | | *Ethanoligenens* | 0.96 |
| | | *Monoglobus* | 0.79 |

to the species *Marvinbryantia formatexigens*, which has been found in the human gut and has been reported to degrade plant oligosaccharides (stachyose and raffinose) (*Rey et al., 2010*).

## Alpha diversity and microbial composition

In all samples, bacterial and archaeal phyla and genera showed a sample coverage of 100%, indicating that sampling is complete for all samples (Fig. 3). Then the diversity comparisons were made directly according to their confidence intervals (*Chao & Jost, 2012*). For bacterial communities, the richness of S1 was significantly higher than in the other samples, at both phylum (Fig. 4A) and genus level (Fig. 4C). For the phyla of Archaea domain, the richness of both S1 and S2 were not significantly different while a similar pattern was observed between S3 and S4 (Fig. 4B). Richness estimates at the genus level in both domains differed significantly between all samples (Fig. 4C; Fig. 4D). The diversity values of the effective number of common and dominant elements are shown in Table S2.

According to ANOSIM, community composition at Bacteria and Archaea domains was not significantly different between samples ($p > 0.05$). For genera of domain Bacteria, the composition was significantly different between S1 and the other three samples ($p = 0.0002$), while for the genera of domain Archaea the composition was significantly different between all samples ($p = 0.0002$).

## Functional annotation

A similar number of sequences was annotated from all samples (101,832–138,018) in all categories (Table 2). According to the function of their specific catalytic action, genes encoded oxidoreductases, transferases, hydrolases, lyases, isomerases, ligases, and translocases (*Chang et al., 2021*). The most abundant genes in all samples were transferases (28,523–21,324 sequences) and hydrolases (17,885–24,675) followed by oxidoreductases (10,797-7,757), ligases (5,081-7,013), lyases (4,896–6,064) and isomerases (4,227-5,500), no translocase-type enzymes were found with the PROKKA annotation (Fig. 5). The highest number of annotations of all the enzymes (82,572) was in S1 while the sample with the lowest number of annotated enzymes was S2 (61,201) (Fig. 5). Additional enzyme search using blastp showed between 954–1,125 total in different samples. Genes encoding translocases were recovered, which ranged from 16 to 37 sequences in the different samples. We found 21 functional categories. Protein Families (17,065–11,711 contigs per sample), genetic information processing (17,065–12,536), carbohydrate metabolism
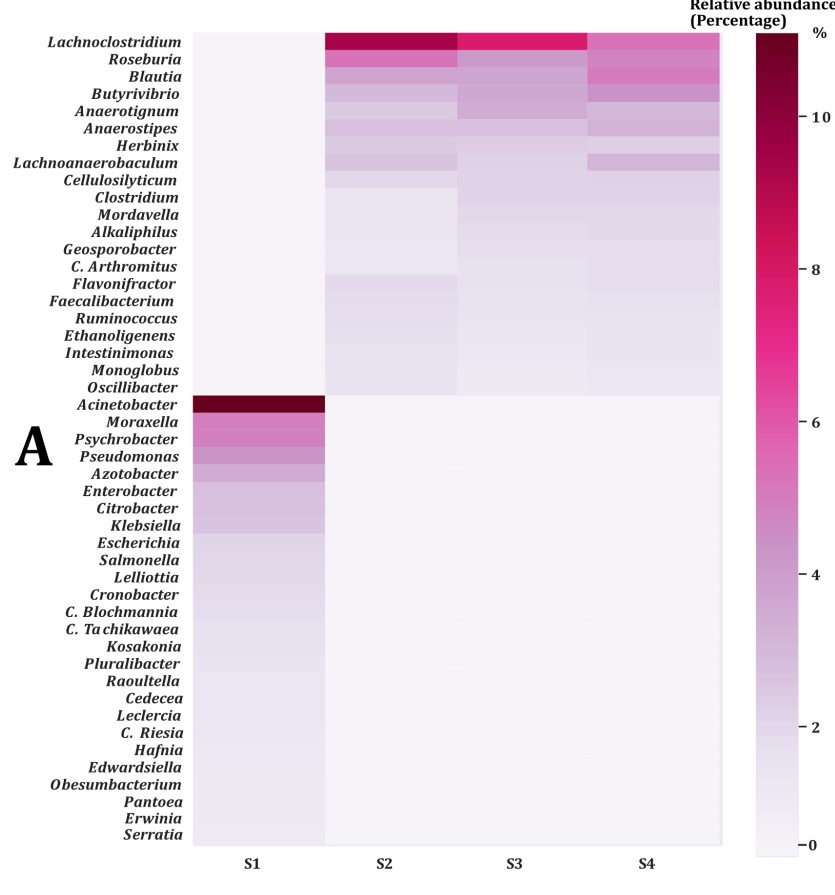

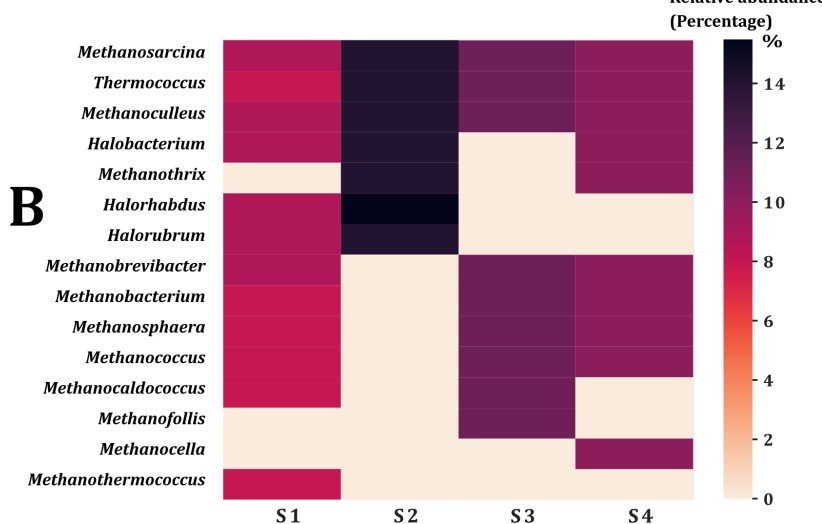

**Figure 2  Genera of the bacteria and archaea domain found in the fecal microbiome of the volcano rabbit.** Relative abundance was obtained by dividing the number of contigs assigned per genus by the total number of contigs for the Bacteria and Archaea domains, respectively. The figures were constructed considering only genera with a relative abundance equal to or greater than 0.5%, which means that rare bacteria and archaea were not considered. (A) Percentage of abundance of the most representative genera per sample within Bacteria domain. (B) Percentage of abundance of the most representative genera per sample within Archaea domain.

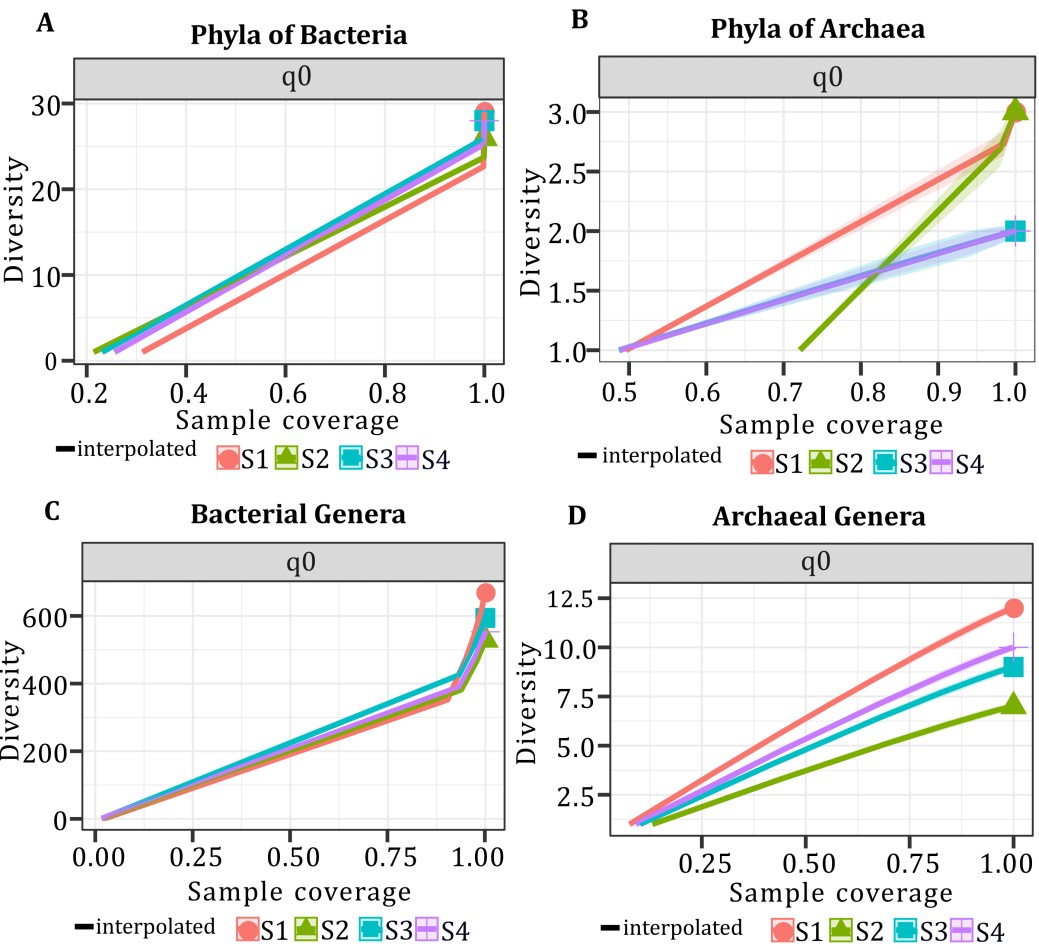

**Figure 3** **Sample coverage (SC) per sample, taxonomic level, and diversity (q0).** SC ranged from 0.0 to 1.0. (A) Sample coverage for the phyla of the Bacteria domain. (B) Sample coverage for the phyla of Archaea domain. (C) Sample coverage for the genera of the Bacteria domain. (D) Sample coverage for the genera of the Archaea domain.

(glycolysis/gluconeogenesis, citrate cycle, TCA cycle; 12,571–9,776), protein families: signaling and cellular processes (7,400–5,197) and amino acid metabolism (7,103–5,730) were the most abundant in all samples. Meanwhile, xenobiotics biodegradation (127–58) and biosynthesis of secondary metabolites (49–66) had the lowest number of annotated sequences (Fig. 6).

Tannase enzyme sequences from different taxonomic groups were searched and downloaded from UniProtKB/Swiss-Prot and NCBI databases. Blastp search was performed to obtain the possible gene sequences for tannase enzymes. 18 sequences were found in the four metagenomic samples of volcano rabbits that were affiliated to phylum Verrucomicrobia, family Ruminococcaceae, *Victrivallis vadensis*, *Blautia* sp. and *Clostridium* sp. Finally, enzymes that participate in the degradation of cellulose and hemicellulose of grasses were searched. The highest number of sequences was obtained from young rabbit feces (849), while the lowest number was obtained in S2 (526) (Fig.

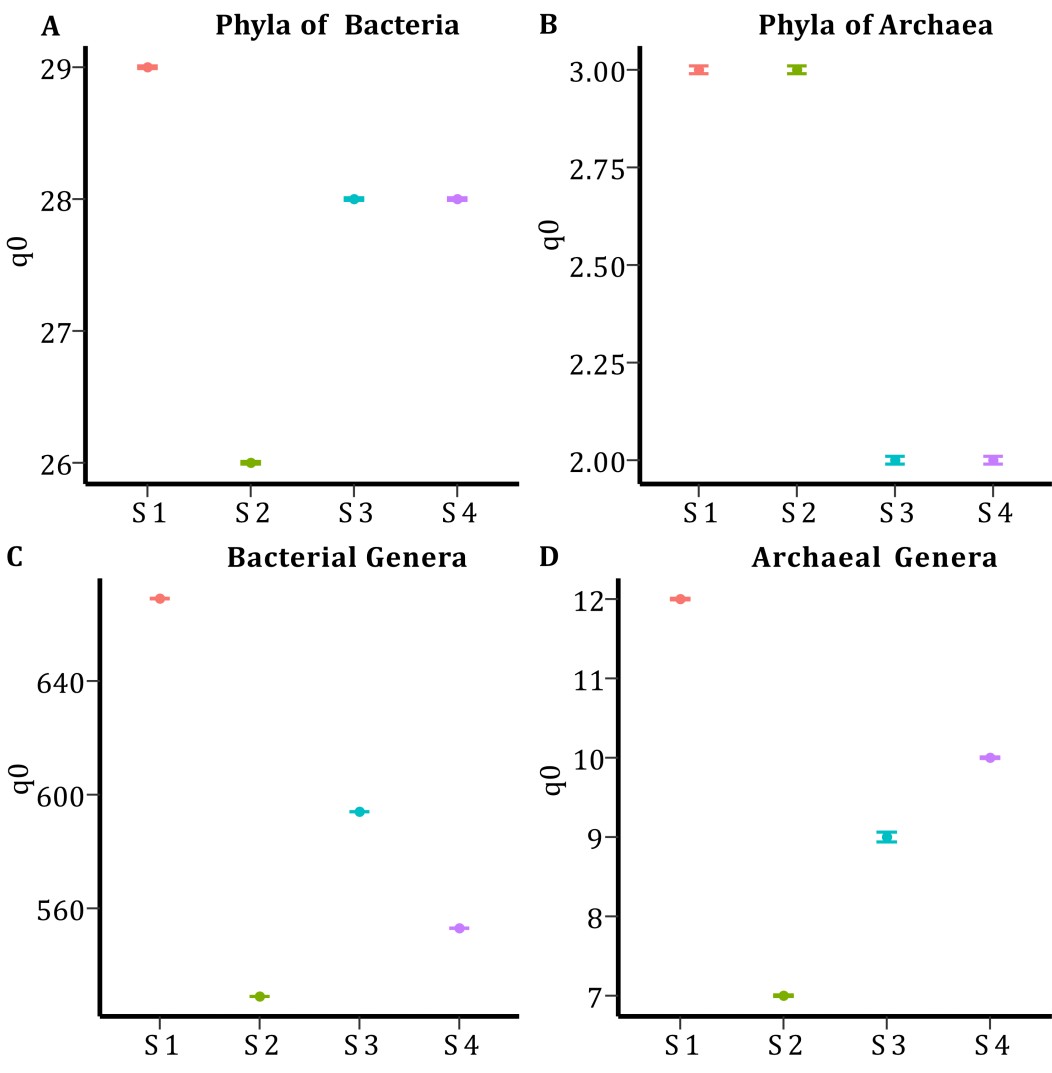

**Figure 4** **Species Richness (q0) per sample and taxonomic level.** (A) Species richness for the phyla of the Bacteria domain. (B) Species richness for the phyla of Archaea domain. (C) Species richness for the genera of the Bacteria domain. (D) Species richness for the genera of the Archaea domain.

**Table 2** **Functional annotation comparison between all samples.** The numbers indicate the total of annotated contigs in each category per sample.

| Categories | S1 | S2 | S3 | S4 |
|---|---|---|---|---|
| CDS | 131028 | 96532 | 101789 | 119898 |
| tmRNA | 90 | 63 | 69 | 87 |
| rRNA | 511 | 387 | 387 | 422 |
| tRNA | 6389 | 4850 | 5098 | 5430 |

7). The most abundant were endo 1,4 $\beta$-xylanases for samples S1 (363) and S4 (248) and arabinofuranosidases for samples S2 (160) and S3 (149). The enzymes found in lower

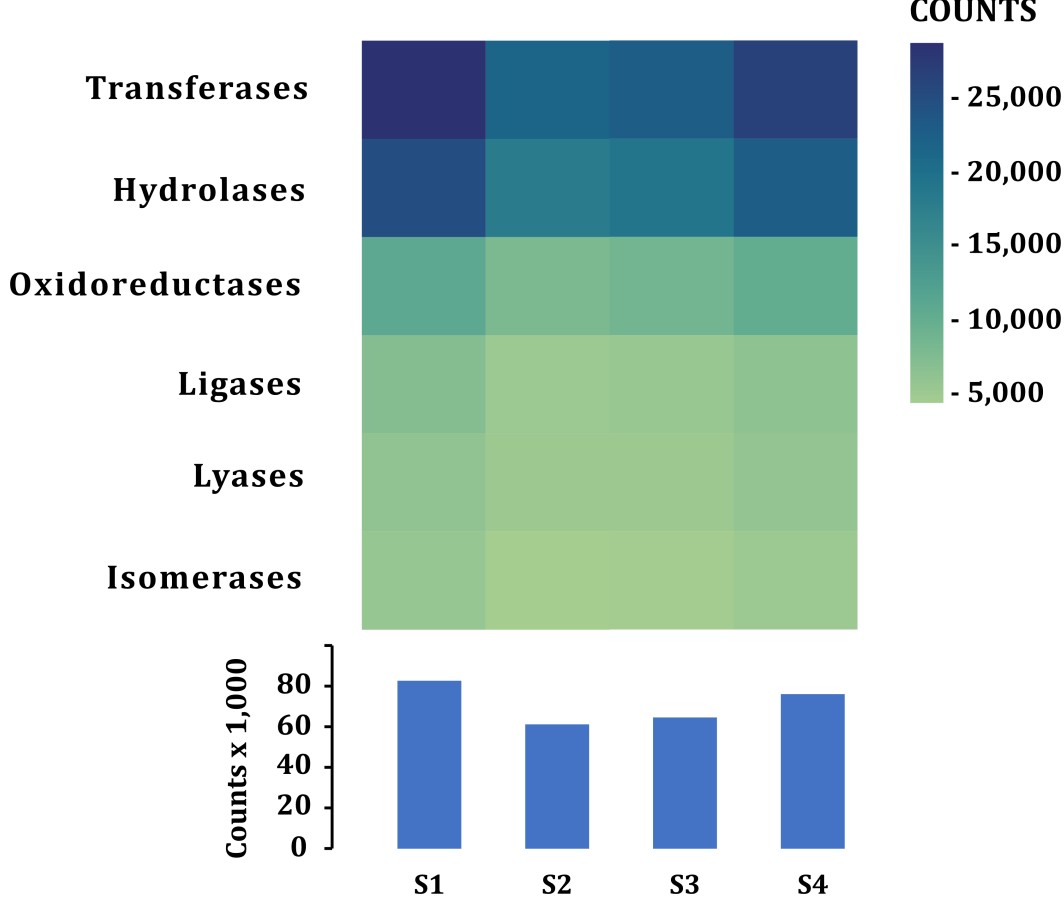

**Figure 5 Enzymes annotated in the fecal metagenome of the volcano rabbit.** Comparison of enzymes according to their catalytic action: Transferases, Hydrolases, Oxidoreductases, Ligases, Lyases and Isomerases per sample and by number of contigs (COUNTS) using PROKKA.

proportion were those encoding $\beta$-glucosidases in sample S2 (94) and endoglucanases in samples S3 (142) (Fig. 7).

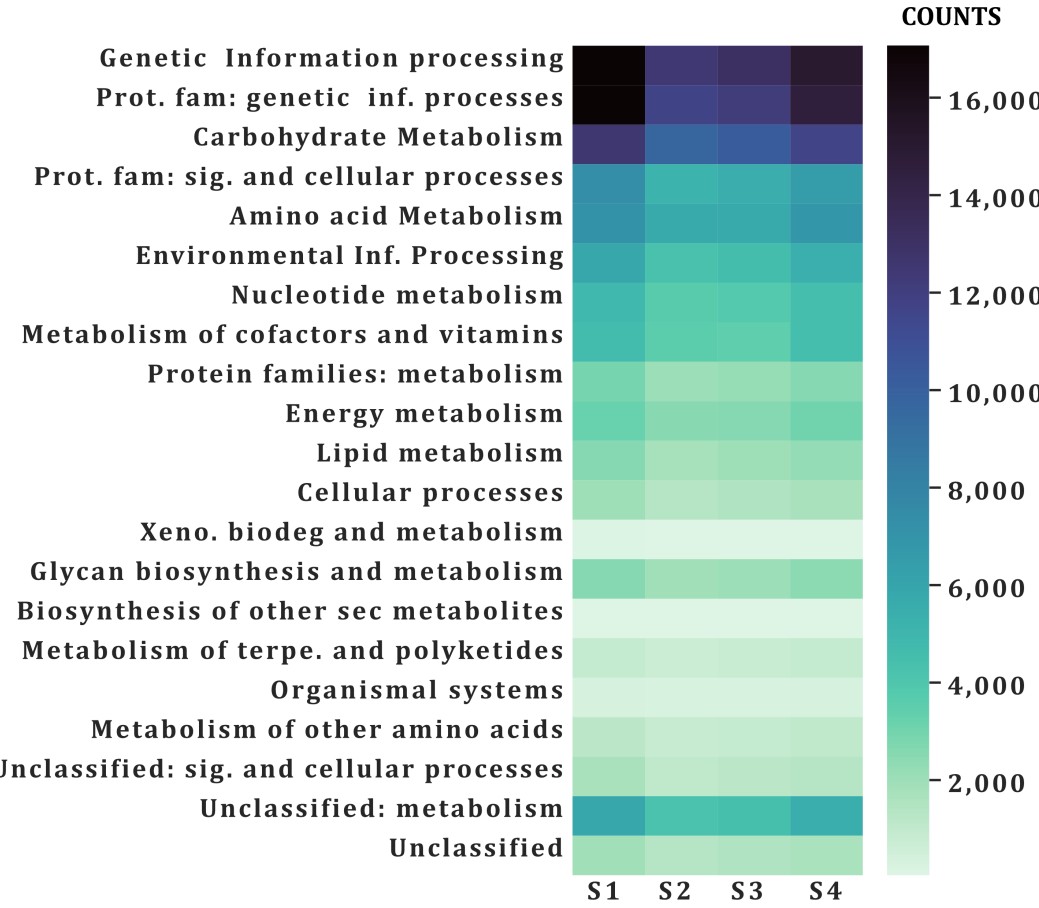

**Figure 6** **Functional annotation categories in the fecal metagenome of the volcano rabbit.** Comparison of categories of functional annotation by sample and by number of contigs (COUNTS) using GhostKOALA.

## DISCUSSION

### Gut microbiota

There are few studies on the microbiota of wild Lagomorphs. To our knowledge, this is the first one to characterize the fecal microbiome of the volcano rabbit (*R. diazi*) that is endemic to central highlands in Mexico. As an herbivore, the volcano rabbit would feed on plants that contain endophytes which may become part of the gut microbiota. This occurs in other herbivorous animals' endophytes that have tannases and enzymes to degrade plant cell wall components (*Martinez-Romero et al., 2021*). A critical issue that should be considered is whether the microbiota identified from herbivorous animals represent *bona fide* gut bacteria or reflect endophytes that are still contained within the plant tissue fragments macerated during DNA extraction. Here a mild procedure not including physical maceration was performed and all plant debris was eliminated by centrifugation.

Diet changes may have fast and important consequences in gut microbiota (*Human Microbiome Project Consortium, 2012*). The uncontrolled diets of *R. diazi* in their natural

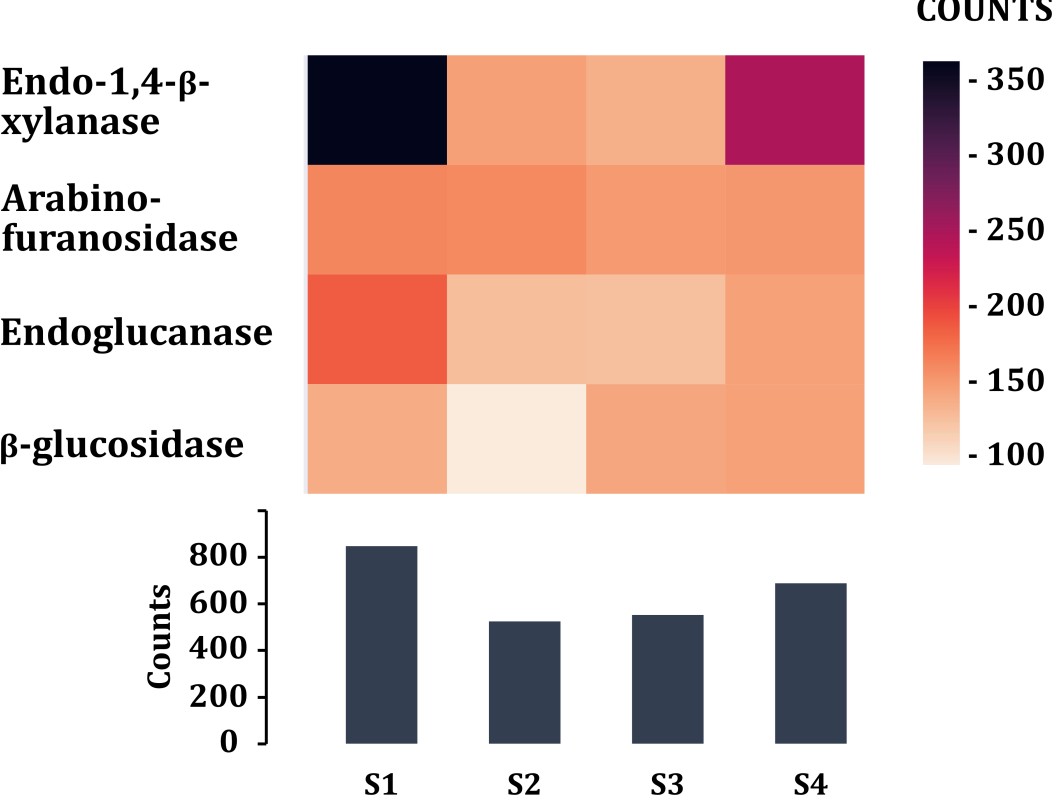

**Figure 7** **Enzymes (class 3) associated in the degradation of cellulose and hemicellulose of plants.** Comparison of the enzymes associated with plant fiber degradation by sample and by number of contigs (COUNTS).

habitat may explain differences among adult samples. Otherwise, differences could be due to the distinct lapse from fecal excretion and collection for which we have no record. Peculiarities of the gut microbiota of babies, infants and young animals have been recurrently reported, and here we corroborated this with the only young rabbit that was sampled. There are reports from three-week-old Asian elephants (*Loxodonta cyclotis*) (*Ilmberger et al., 2014*), young bats (*Leptonycteris yerbabuenae*) (*Gaona et al., 2019*) and puppies and kittens (*Moon et al., 2018*) where Proteobacteria are abundant. Since the gut of newborns is rich in oxygen, the role of Proteobacteria may be involved in oxygen consumption, thus preparing the neonatal gut for colonization by strict anaerobic microorganisms necessary for a healthy gut (*Shin, Whon & Bae, 2015*).

Here we report that the fecal microbiome of the volcano rabbit was dominated by Proteobacteria, Firmicutes, Actinobacteria, for S1 and S2 and Firmicutes, Actinobacteria and Cyanobacteria for S3 and S4. We suggest that the similarity between S1 and S2 may be due to the individuals being related because young rabbits tend to be close to their mothers and these samples were collected at the same location. However, we remark that in zoos and in houses, rabbits would have an artificial diet. It is also worth noting that fecal and gut bacteria are not the same and more Proteobacteria were found in intestinal samples

than in fecal samples in the European brown hare (*Stalder et al., 2019*). Additionally, Proteobacteria has been reported to be more abundant in rabbits with the symptoms of epizootic rabbit enteropathy (ERE) (*Bauerl et al., 2014*) and in other mammals, such as bats (*Artibeus lituratus*) (*Ingala et al., 2018*) and Asian elephants with a high fruit diet (*Budd et al., 2020*). On the other hand, a study in New Zealand white rabbits has reported a high abundance of Actinobacteria and Tenericutes (*North, Dalle Zotte & Hoffman, 2019*). Phylum Lentisphaerae was previously reported in hares (*L. europaeus*).

On the other hand, 16S rRNA reads from shotgun metagenomics provide sequences that are not subject to PCR primer bias and covers taxa that are not detected by primer sets (*Yuan et al., 2015*); therefore, we complemented the kraken 2 taxonomic assignment with a global 16S rRNA gene bacterial identification and phylogeny from metagenomes. We obtained only nine phyla in contrast to the 29 found with the kraken 2 analysis. Phylum Firmicutes was the most abundant in both. It has been reported that kraken 2 is suitable for the classification of gut microbiome with assemblies' recruit of 85% or more of the original raw reads (*Tamames, Cobo-Simon & Puente-Sanchez, 2019*). Nevertheless, samples that have more diversity and complexity would not be correctly classified with kraken 2. Less studied habitat metagenomes such as non-human environments show fewer similarities with kraken 2 which is quite sensitive to the composition of databases and their performance decreases when rare species are present in the metagenome (*Tamames, Cobo-Simon & Puente-Sanchez, 2019*).

Domain Archaea has been reported in Caldes rabbit microbiome (*Velasco-Galilea et al., 2018*) with a single genus *Methanobrevibacter* (Euryarchaeota). Similarly, genus *Methanobrevibacter* was identified in a molecular profiling from a rabbit caecum (*Kusar & Avgustin, 2010*). These differences could be due to different strategies to characterize microbiota (16S rRNA amplicon sequencing *vs* shotgun). Here we found an unusual high number (15) of archaeal genera in the fecal microbiome of the volcano rabbit. Surprisingly, the most abundant genera were *Methanosarcina*, *Methanoculleus*, *Methanococcus*, *Halorhabdus* (Euryarchaeota) besides *Methanobrevibacter*. All of them have been found in the human gut microbiome (*Borrel et al., 2020*). *Halorhabdus* is usually associated with microaerophilic and halophilic environments. The axenic species described may use the monosaccharides glucose, fructose, and xylose for growth (*Waino, Tindall & Ingvorsen, 2000*). Xylose is produced mainly in plants (wood, fibers, fruits) and as a degradation product of hemicellulose. Except for *Halorhabdus*, the other archaea may perform the terminal step in the degradation of organic matter and produce energy and methane through methanogenesis pathways (*Ferrer & Pérez, 2010*; *Ferry, 2020*). These methanogenic archaea are generally associated with strict anaerobic environments, due to the high sensitivity to oxygen exposure of the enzymes involved in methanogenesis and to the lack of genes that code for proteins involved in protection against oxidative stress (*Jasso-Chavez et al., 2015*). Furthermore, these methanogenic archaea use compounds with one or two carbons to grow, for the generation of biosynthetic intermediates and polysaccharide storage (*Santiago-Martinez et al., 2016*). *Methanoculleus*, *Methanococcus* and *Methanobrevibacter* use carbon dioxide plus hydrogen ($CO_2 + H_2$) and formate (HCOO-), while *Methanosarcina* uses more diverse carbon sources, *e.g.*, acetate ($CH_3COO-$),

methanol ($CH_3OH$) and methylamines ($CH_3NH_2$), plus $CO_2 + H_2$ (*Borrel et al., 2020*; *Buan & Robinson, 2018*). These compounds are produced by the metabolism of eukaryotic and bacterial species. Specifically, there is evidence that the high intake of dietary fiber in animals, followed by fermentation mediated by the intestinal microbiota, changes the concentration and metabolism of methylamines (*Li et al., 2017*), which are a very good carbon source for methanogenic archaea. This suggests that the type of diet could increase the diversity and abundance of methanogenic species. The diversity and abundance of methanogenic archaea found in volcano rabbits may indicate a rich diversity of ingested vegetable and very complex interaction between bacteria and archaea. Seemingly there is no competition for food between these methanogenic and non-methanogenic archaea, due to the well-established difference in preference for carbon sources.

## Prediction of functional annotation

In the present study, we found several functional categories suggesting that the fecal microbiome of *R. diazi* may have biosynthetic amino acid, carbohydrate metabolism, lipid metabolism, metabolism of terpenoids, polyketides and secondary metabolites and xenobiotic biodegradation capabilities. Enzymes found in the fecal microbiome of the volcano rabbit were mainly transferases, hydrolases, and oxidoreductases enzymes. Categories such as carbohydrate metabolism and amino acid metabolism had been reported in other herbivores (*Muegge et al., 2011*). Pathways such as bile secretion, mineral absorption, and xenobiotic biodegradation have been obtained in other rabbits (*Zeng et al., 2015*). Recently, metabolism of terpenoids –polyketides, amino acid metabolism, carbohydrate metabolism, nucleotide metabolism, lipid metabolism, energy metabolism, and metabolism of cofactors and vitamins pathways were reported in African elephants (*Budd et al., 2020*).

Enzymatic activity of tannases was identified in phyla Firmicutes and Proteobacteria: *Enterobacter*, *Weissella* and *Lactobacillus* (de las (*de las Rivas et al., 2019*), *Bacillus*, *Streptococcus*, *Klebsiella*, *Enterococcus fecalis*, *Pantoea agglomerans*, *Staphylococcus lugdunensis*, *Lactobacillus plantarum*, *L. paraplantarum* (*Rodríguez-Durán et al., 2010*) and *Bacillus licheniformis* (*Palacio-Arango et al., 2018*). In Koala feces, a Gram-negative, facultatively anaerobic and tannase-producing bacterium called *Lonepinella koalarum* was isolated (*Osawa et al., 1995*) which was not found in the microbiome of volcano rabbit feces pointing out the specialization or adaptation of some bacterial species to animal hosts.

Fiber degrading enzymes have been reported in bacteria such as *Clostridium*, *Cellulomonas*, *Bacillus*, *Thermomonospora*, *Ruminococcus*, *Bacteroides*, *Erwinia* and *Streptomyces* (*Saratale, Saratale & Oh, 2012*). *Clostridium* was abundant in S2, S3 and S4 whereas *Erwinia* in S1. In all samples, we also found phylum Actinobacteria, which is abundant in soil producing secondary metabolites, antibiotics and related with degrading plant cell wall components (*Saratale, Saratale & Oh, 2012*).

## CONCLUSIONS

The volcano rabbit microbiome showed distinct bacterial and archaea abundances compared to other lagomorphs. We found genes that encode tannases and enzymes

that degrade the components of the plant cell wall in the volcano rabbit microbiome. The gut microbiota may contribute to the digestion of complex plant molecules. Here we found many potential functional categories such as metabolism of carbohydrates and biosynthesis of amino acids and other compounds as secondary metabolites. The diversity of methanogenic species could be influenced by the type of diet. In addition, we observed differences between the adults and young rabbit.

## ACKNOWLEDGEMENTS

We thank all the colleagues from the FCB-UAEM for their help in the field work as well as Biol. Tonalli García, M.S. Diana Oaxaca and M.S. Víctor Higareda from CCG-UNAM for their advice in bioinformatics tools. Dr. Julio Guerrero from Umea University for his assistance in the molecular assays. Dr. Lydia Smith from UC Berkeley for contacting us with the sequencing unit. M.S. Gustavo Delgado from IBT-UNAM for his assistance and advice with phylogeny and bioinformatics tools. Dr. Geovanni Santiago from Pennsylvania State University for his helpful comments on archaea. Dr Mónica Rosenblueth from CCG-UNAM for her helpful comments on the manuscript. Dr M. Dunn from CCG-UNAM for critically reading the manuscript. We thank the CCG-UNAM for giving us access to its computer cluster.

### Funding

This study was performed with financial support by a grant from the Comision Nacional de Areas Naturales Protegidas (CONANP) through the Project PROMANP/MB/50/ 2018 to Jose Antonio Guerrero and a grant from PAPIIT UNAM IN200021 to Esperanza Martinez Romero. Leslie Mariella Montes Carreto is a doctoral student at Programa de Doctorado en Ciencias Naturales of the Universidad Autónoma del Estado de Morelos (UAEM), with a scholarship (no. 291236, CVU: 667266) from Consejo Nacional de Ciencia y TecnologÃa (CONACyT). Support was also received by Comisión de Operación y Fomento de Actividades Académicas (COFAA) and Instituto Politécnico Nacional (IPN). The funders had no role in study design, data collection and analysis, decision to publish, or preparation of the manuscript.

### Grant Disclosures

The following grant information was disclosed by the authors:
Comision Nacional de Areas Naturales Protegidas (CONANP): PROMANP/MB/50/ 2018.
PAPIIT UNAM: IN200021.
Universidad Autónoma del Estado de Morelos (UAEM), with a scholarship: 291236, CVU: 667266.
Consejo Nacional de Ciencia y Tecnología (CONACyT).

### Competing Interests

The authors declare there are no competing interests.

## Author Contributions

- Leslie M. Montes-Carreto conceived and designed the experiments, performed the experiments, analyzed the data, prepared figures and/or tables, authored or reviewed drafts of the paper, and approved the final draft.
- José Luis Aguirre-Noyola performed the experiments, authored and reviewed drafts of the paper, and approved the final draft.
- Itzel A. Solís-García analyzed the data, prepared figures and/or tables, authored or reviewed drafts of the paper, and approved the final draft.
- Jorge Ortega conceived and designed the experiments, authored or reviewed drafts of the paper, and approved the final draft.
- Esperanza Martinez-Romero conceived and designed the experiments, analyzed the data, authored and reviewed drafts of the paper, and approved the final draft.
- José Antonio Guerrero conceived and designed the experiments, analyzed the data, prepared figures and/or tables, authored or reviewed drafts of the paper, and approved the final draft.

## Field Study Permissions

The following information was supplied relating to field study approvals (i.e., approving body and any reference numbers):

Secretaría de Medio Ambiente y Recursos Naturales approved this research (SGPA/DGVS/006985/18).

## Data Availability

The data are available in the SRA NCBI, BioProject:

PRJNA721235, SRR14209496, SRR14209495, SRR14209494, SRR14209493.

## Supplemental Information

Supplemental information for this article can be found online at http://dx.doi.org/10.7717/peerj.11942#supplemental-information.

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
