# Peer review of "Diverse methanogens, bacteria and tannase genes in the feces of the endangered volcano rabbit (Romerolagus diazi)"

_PeerJ, doi:10.7717/peerj.11942_

## Round 0.1 · original submission · Minor Revisions

I have received two reviews and they are positive in general. I think the main issue, and something you need to account for, is the small sample size. I understand that collecting these types of samples may be difficult. However, at the very least, you should include a statistical power analysis for microbiome data and a discussion of the potential impact of examining such few samples. Ideally, you should include more samples to make your results more robust. By statistical power analysis, I'm looking for the probability of finding a taxon or a gene, or by detecting differential abundance of taxa. What's the impact of sample size on finding methanogens and genes of interest?

Reviewer 1 ·

Basic reporting

This paper by Montes-Carreto et al investigated the faecal microbiome by metagenomic sequencing from four faecal pellets thought to be from the endangered Volcano rabbit in Mexico.

Although I originally accepted this article for peer review, on reading the article I do not feel that I have appropriate expertise to review this paper. I have expertise in some lagomorphs, but not the ecology of the volcano rabbit. I have some expertise in metatranscriptomics for virus discovery. But I feel that the bacterial ecology and bioinformatics used in this manuscript go beyond my area of expertise.

I have tried to provide some useful comments below but I really struggled to understand the manuscript, analyses, and conclusions, presumably because of my lack of expertise specifically in bacterial ecology.

Experimental design

Based on my experience with metatranscriptomic sequencing, I would not typically consider an analysis of four animals to be robust. I do not know if this is acceptable in the microbial ecology field. I would also have concerns about assigning animal age based on faecal size, assigning species based on faecal appearance, cross-contamination with two samples being collected from the same latrine, representativeness of faecal samples that were not collected fresh (e.g. were samples degraded due to environmental/UV exposure?).
Based on my experiences, I have some concerns about the external validity of these findings and extrapolation from the small sample size. For example, the authors make conclusions about all young volcano rabbits, and differences between young and adult volcano rabbits, based on a single faecal sample that may be from a young animal based on faecal pellet size.

Validity of the findings

I had difficulties assessing the validity of the findings, likely due to my lack of specific expertise. This paper should be reviewed by an additional reviewer with more microbial shotgun bacterial metagenomics expertise.

However, I did note some inconsistencies in the manuscript.
For example, Lines 86-88 – The author's interpretation and summary of these papers is not how I read these papers. Velasco-Galilea found that Firmicutes, followed by Bacteroidetes and Tenericutes, were the most abundant phyla. Crowley found that Firmicutes and Bacteroidetes were the most abundant. North found that Firmicutes, followed by Bacteroidetes and Proteobacteria were the most abundant but they also looked at caecal samples, which may differ from faecal samples investigated in many of the other papers. Shanmuganandam found that Illumina 16S v3-v4 sequencing underrepresented Tenericutes.
For example, in Figure S1 I would typically look for a more comprehensive Leporidae tree.
For example, I interpreted figure S2 as not supporting Figure 1 e.g. I could not see any Actinobacteria in Fig S2. However, the authors state in line 315 that the same 9 phyla were found by 16S and shotgun analysis. I am not sure how much correlation there should be between shotgun and 16S datasets.

Additional comments

After reviewing this article I found that I did not have the relevant expertise to review this appropriately. This paper should be reviewed by an additional reviewer with more microbial shotgun bacterial metagenomics expertise. Since I work mostly with viruses, I can't really think of anyone with specific bacterial metagenomics expertise. I'm sorry that I could not be of more help.

Reviewer 2 ·

Basic reporting

The English language should be reviewed to ensure that an international audience can clearly understand your text. I suggest you contact a native colleague or a professional service that help you to improve the English of your manuscript.

Experimental design

Your research aims at characterizing the fecal microbiota of the volcano rabbit and assessing the role of microorganisms in the degradation of plant fiber and tannin. I found it a meaningful objective since you sed light on a Mexican endemic rabbit species, which, moreover, is critically endangered. However, you should remove the last sentence of the introduction section "Indeed, we found abundant genes with such putative functions, especially endoglucanase genes, but also β-glucosidases, arabinofuranosidase and endo 1,4 β-xylanase." because this is an output of your study.

My main concern in your study is its limited sample size. This constraint, however, could be justified by the difficulties in collecting feces of an endangered species. Moreover, this study is of interest because it covers a topic in a species which is still relatively poorly understood.

Validity of the findings

The limited number of samples is the main limitation of this study, specially in terms of providing results statistically sound.

Additional comments

I think microbial names should be italicised.

L30: citations should not be provided in the abstract.
L57: it should be "interacting between them".
L58: remove "or disease".
L63-65: please add references of previous studies in rabbits that demonstrate how age, genetics, diet, breeding conditions or the administration of antibiotics shape gut microbiota over the life of the animal.
L74: which species?
L87: "dominated by phyla..."
L103: please explain the importance of finding microorganisms involved in tannin degradation.
L119: how much time were samples stored until DNA extraction?
L127: please provide more detail about the process of library generation.
L170: how did you compute Shannon index? Specify at which number of reads you rarified your samples.
L175: please explain why you used nonparametric Friedman test with 10,000 Bootstrap.
L178: did you conduct any additional filtering by low abundant genera?
L201: please specy the range and the average number of raw and filtered reads per sample.
L209: At which taxonomy level does it refer? please provide an explanation for this low percentage of classification.
L212: you should provide some descriptive statistics in this table.
L323: you should mention that these differences could be due to different strategies to characterize microbiota (shotgun vs 16S amplicon sequencing).

---

## Round 0.2 · accepted · Accept

I'd like to thank the authors for taking care of the reviewers' comments. I think the manuscript is in better shape now and I'm happy to accept it for publication.